# Probabilistic Prediction of Strength and Fracture Toughness Scatters for Ceramics Using Normal Distribution

**DOI:** 10.3390/ma12050727

**Published:** 2019-03-02

**Authors:** Chunguo Zhang, Shuangge Yang

**Affiliations:** Key Laboratory of Road Construction Technology and Equipment, MOE, Chang’an University, Xi’an 710064, China; ysg2307105477@163.com

**Keywords:** tensile strength, fracture toughness, average grain size, normal distribution, ceramic

## Abstract

Tensile strength *f*_t_ and fracture toughness *K*_IC_ of ceramic are not deterministic properties or fixed values, but fluctuate within certain ranges. A nonlinear elastic fracture mechanics model was developed in this study and combined with the common normal distribution to predict ceramic’s *f*_t_ and *K*_IC_ with consideration of their scatters in a statistical sense. In the model, the relative characteristic crack size *a**_ch_/*G* (characteristic crack size *a**_ch_, average grain size *G*) was determined based on the fracture measurements on five types of ceramics with different *G* from 2 to 20 μm in the reference (Usami S, et al., Eng. Fract Mech. 1986, 23, 745). The combined application of the model and normal distribution has two functions: (i) probabilistic *f*_t_ and *K*_IC_ can be derived from seemingly randomly varied fracture tests on small ceramic specimens containing different initial defects/cracks, and (ii) with *f*_t_ or *K*_IC_ values (corresponding mean and standard deviation), fracture strength of heterogeneous samples with and without cracks can be predicted by considering scatter described by specified reliability. For the fine ceramics, the predicted results containing the mean and the upper and lower bounds with 96% reliability gained with the model, match very well with the experimental results (*a*, *σ*_N_).

## 1. Introduction

One of the biggest challenges in material sensitive design is to accurately and reliably predict the fundamental mechanical properties of materials. However, the physical properties such as tensile strength *f*_t_ and fracture toughness *K*_IC_ are not fixed or deterministic, and fluctuate in certain range, which has been well recognized recently [1,2,3]. 

The inevitable variation in mechanical property of materials is mainly due to the heterogeneity of microstructure, defects like pores and the activation of various fracture mechanisms during the crack-microstructure interactions [4]. For polycrystalline brittle materials such as fine-grained ceramics [5,6,7] and coarse-grained rocks [8,9,10], the average grain size *G* has a profound influence on material properties including *f*_t_ and *K*_IC_. For composites and 3D-printed scaffolds [11], their unit cells have a similar role as that of *G*. 

Nonlinear elastic fracture of those solids initiated from shallow defects comparable to their grain sizes can bring out significant information on the inherent relationship between microstructure and material properties. Besides separate influence on *f*_t_ and *K*_IC_, the effect of *G* on the combined parameter 0.25*(*K*_IC_/*f*_t_)^2^, commonly known as characteristic crack size *a**_ch_ indicating the transition from *f*_t_-controlled to *K*_IC_-controlled fracture of brittle solid [12,13], is becoming increasingly important. Taylor has done numerous studies on characteristic length *l*_ch_, proportional to *a**_ch_, and proposed the world-renowned theory of critical distance (TCD) [14,15,16,17,18,19]. It has been found that the *l*_ch_ varies from 1 to 10 times of grain size *G* depending on the materials. Generally speaking, *l*_ch_ = (3–4)*G* can be adopted for various fine-grained ceramics. More recently, we reanalysed the fracture measurements data on four fine-grained ceramics with *G* from 2 to 20 μm and on two coarse-grained rocks with *G* = 2.5 and 10 mm, and also found a semi-quantitative relation of *a**_ch_ ≥ 3*G* [7,10].Therefore, an explicit relationship between *a**_ch_ and *G* is an urgency need not only for the fundamental knowledge of microstructure-driven material research, but also for the safe design application.

For fracture data analysis methods, the common practice is that fracture strength is first obtained from experiment and then fitted by various fracture models, and the scatters are usually indicated by the maximum and minimum values but rarely described with a specified reliability [20,21,22,23,24,25,26]. The problem is that the models cannot describe the scatter of physical properties or fracture strength with specified reliability. In addition, many parameters in the models are fitting parameters which carry little physical significance. 

In this study, five groups of fracture measurements on ceramics specimens with *G* from 2 to 20 μm and containing different initial defects from 100 nm to 800 μm [27,28] were reanalyzed to determine the relative characteristic crack size *C* = *a**_ch_/*G*. Based on this, a non- linear elastic fracture mechanics (non-LEFM) analytical model is proposed to predict *f*_t_ and *K*_IC_ of ceramics as function of grain size *G* from common fracture measurements, and to predict fracture strength of specimens with and without small defects if *f*_t_ or *K*_IC_ are available. Instead of curve fitting, the common normal distribution was used to analyse the fracture behaviour of ceramics to consider the scatters of physical and mechanical properties. 

## 2. Determination of the Relative Characteristic Crack Size in Non-LEFM Model

Based on our previous work [7], fracture strength of ceramic specimens containing small flaws/cracks can be assessed by the following formula,
(1a)$$\sigma_{N}=f_{t}\cdot \frac{1}{\sqrt{1+\frac{a}{C\cdot G}}}$$
where the combined parameter *C*⋅*G* = *a**_ch_ = 0.25(*K*_IC_/*f*_t_)^2^ is a material constant defined by the intersection of two fracture criteria *K*_IC_ and *f*_t_, and *σ*_N_ is the engineering stress without consideration of the influence of initial crack size *a*.

Although Weibull distribution has been widely used to determine the fracture behavior of materials, its role has usually been confined to fit experimental data rather than providing predictive insight into material properties (e.g., *f*_t_ and *K*_IC_) and their experimental scatters [4]. In our previous study [7,10], another common statistical analysis method of normal distribution has been used to analyze experimental data of ceramic fracture after roughly assessing the relative characteristic crack size *C* (= *a**_ch_/*G*) at intervals of 0.25. In this study, we accurately determine the *C* value by continuously changing its value instead of interval.

The average grain size *G* is the microstructure characteristic of a polycrystalline material which can be statistically measured separately.If the *C* is a fixed value, Equation (1a) can be linearized by considering *σ*_N_ and 1/[1 + *a*/(*C*⋅*G*)]^0.5^ as ordinate and abscissa respectively, then it can be used in conjunction with the common normal distribution to make probabilistic analysis on *f*_t_ with 96% reliability. That is, if an experimental data (*σ*_N_, *a*) is measured, the *f*_t_ from Equation (1a) becomes a single parameter after giving *C* a fixed value. Having numerous pairs of values of original crack *a* and engineering stress *σ*_N_ obtained from specimens with different α ratio (= *a*/*W*, *W* is the beam/specimen height), the data points (*σ*_N_, 1/[1 + *a*/(*C*⋅*G*)]^0.5^) obtained can be plotted with these quantities considered as ordinate and abscissa respectively. Using Equation (1a), a group of *f*_t_ values can be easily statistically analysed by normal distribution mehthodology.

Similarly, the following equation can be obtained by substituing the relation *C*⋅*G* = 0.25(*K*_IC_/*f*_t_)^2^ into Equation (1a), which is used to calculate *K*_IC_ value from fracture measurement (*σ*_N_, *a*). Similarly, Equation (1b) combined with normal distribution can be used to do statistical analyses of *K*_IC_ values from experimental measurements data if the *C* is a fixed value.
(1b)$$\sigma_{N}=K_{IC}\cdot \frac{1}{2\sqrt{a+C\cdot G}}$$

Therefore, a determined value of *C* urgently needs to be included in the two formulas of Equation (1). For this reason, the ralative characteristic crack size*C* (= *a**_ch_/*G*) independent of grain size *G* is assessed based on the experimental data (*σ*_N_, *a*) on five ceramics with different *G* from 2 to 20 μm. In total, 14 experimental data points (*σ*_N_, *a*) of the Sialon with average grain size *G* = 2 μm, 29 points of the SiC with *G* = 3 μm, 25 points of Si_3_N_4_ with *G* = 3 μm, 16 points of Si_3_N_4_ with *G* = 4 μm, and 42 points of Al_2_O_3_ with *G* = 20 μm, were digitized from the reference [28]. 

Based on the fracture measurements (*σ*_N_, *a*) of the five ceramics considered in this study, each group of *f*_t_ values from Equation (1a) and *K*_IC_ values from Equation (1b) was analyzed using two ways respectively to ensure reliability of analyzed results. 

Figure 1 illustrates the evolution of standard deviation *σ_f_* of tensile strength *f*_t_ from Equation (1) as the relative characteristic crack size *C* (= *a**_ch_/*G*) varies continuously. *C*_0_ is defined as the critical value of *C* signifying the *σ_f_* transitioned from decrease to increase. That is the *C*_0_ value is determined by the condition that the *σ_f_* value is minimum. When the relative characteristic crack size *C* gradually deviates from the *C*_0_ value, *σ_f_* increases accordingly. According to the analyzed results of the five ceramics with different *G* from 2 to 20 μm, it can be found that *C*_0_ value is not dependent on grain size and fluctuates in a range from 2.8 to 4.7. 

The largest number of experimental data points selected in this analysis is 42 from Al_2_O_3_ fracture, and the minimum number is mere 14 from Sialon. Considering the effect of the fewer number of data points (14 to 42 points) in each group of fracture measurements, we take the average of *C*_0_ values (i.e., 3.380 in this analysis) obtained from the fracture measurements of the five ceramics as the determined value of *C*, which at least should be one of the accepted and reasonable ways. 

For the analysis of *f*_t_ distribution in Figure 1, *f*_t_ is derived from fracture measurement data (*σ*_N_, *a*) through Equation (1a), in which *σ*_N_ and 1/[1 + *a*/(*C*⋅*G*)]^0.5^ follow a linear relation and *f*_t_ in MPa is the slope of the straigth line through the origin. For each data point (*σ*_N_, *a*), the angle between the abscissa and the straight line through the origin and the considered point approaches 90° because the tangent of angle indicating *f*_t_ value is usually around several hundred. Therefore, any tiny error in the degitization of data point (*a*, *σ*_N_) can result in significant fluctuation in the corresponding *f*_t_ value. 

For this reason, we tried another statistic method to determine the *C* value based on the same five groups of ceramic fracture measurements. Figure 2 illustrates the evolution of the sum of distances of all the experimental points to the mean line indicating the average of *f*_t_ values from Equation (1a) as *C* varies continuously. *C*_0_ value is determined by the condition that the sum of distances by normal to the mean line is minimum. It is evident that the sum of the distances gradually increases when the *C* deviates from *C*_0_. It can also be found that *C*_0_ value is not dependent on the grain size *G* and flucturates in a range from 2.6 to 4.3. Again, we took the average of *C*_0_ values obtained from the five different ceramics fracture as the determined *C* value, i.e., 3.1788 in this analysis.

Figure 3 illustrates the evolution of standard deviation *σ*_k_ of *K*_IC_ from Equation (1b) determined by normal distribution as the relative characteristic crack size *C* (= *a**_ch_/*G*) varies continuously for the five ceramics, and the *C*_0_ value is determined by the condition the *σ*_k_ is minimum. The same as above, *C*_0_ value is not dependent on the grain size *G* and flucturates in narrower range from 2.7 to 4.1 in comparison to the results of *f*_t_ distribution. The average of *C*_0_ values obtained from the five different ceramics fracture is 3.1426 in this analysis.

Figure 4 illustrates the evolution of the range of (*K*_IC_)_max_–(*K*_IC_)_min_ from Equation (1b) as the relative characteristic crack size *C* (= *a**_ch_/*G*) varies continuously for the five ceramics, and the *C*_0_ value is determined by the condition the range is minimum. Again, the *C*_0_ value is not dependent on the grain size *G* and flucturates in narrower range from 2.6 to 4.0 in comparison to the results of *f*_t_ distribution. The average of *C*_0_ values obtained from the five different ceramics fracture is 3.1288 in this analysis.

According to the above analyses based on the fracture measurements of the five ceramic with largely different *G*, the relative charateristic crack *C* (= *a**_ch_/*G*) value is indeed not dependent on the grain size *G*, and is eaual to 3.380 and 3.1788 obtained from *f*_t_ distribution, and 3.1426 and 3.1288 from *K*_IC_ distribution. It should be noted that the characteristic crack size *a**_ch_ = *C*⋅*G* is a specified crack indicating the transition from *f*_t_-controlled fracture to *K*_IC_-controlled fracture. Furthermore, the dominant non-LEFM mechanism of brittle materials such as ceramic should have relation to the defects in solids, which may be linked to microstructure characteristic G. If one quantitatively links the specified crack length *a**_ch_ to the material microstructure characteristic *G*, from the perspectives of mathematics and physics the relation of *a**_ch_ = π⋅*G* between the two should be a good choice under condition of *C* ≈ 3.1–3.4 from the above *f*_t_ and *K*_IC_ distribution analyses. In addtion, the *C* values from *K*_IC_ distribution is more stable than those from *f*_t_ distribution, which will be proved in the following section. For simplicity and consistency between *f*_t_ distribution and *K*_IC_ distribution, *C*_0_ = *a**_ch_/*G* = π is selected in this study.

## 3. Probabilistic Strength and Fracture Toughness Analyses

Following the determination of the relative characteristic crack size *C* = π based on the fracture behaviour of five ceramics with *G* from 2 to 20 μm, Equation (1) can now be rewritten as follows.
(2a)$$\sigma_{N}=f_{t}\cdot \frac{1}{\sqrt{1+\frac{a}{\pi \cdot G}}}=f_{t}\cdot \eta \left(a,\;G\right)$$
(2b)$$\sigma_{N}=K_{IC}\cdot \frac{1}{2\sqrt{a+\pi \cdot G}}=K_{IC}\cdot L_{e}\left(a,\;G\right)$$

The dimensionless *η*(*a*, *G*) and equivalent length *L*_e_ (*a*, *G*) in Equation (2) are wholly determined for a specified sample. For any group of ceramic specimens, the seemingly randomly varied fracture measurements data (*σ*_N_, *a*) can be used to obtain a group of *f*_t_ value using Equation (2a) and *K*_IC_ values using Equation (2b) which can be easily analysed by normal distribution to get corresponding mean and standard deviation. With the mean *μ*_f_ and standard deviation *σ*_f_ of *f*_t_ distribution, Equation (2a) can be rewritten to include the mean and upper and lower bounds with 96% reliability indicating *f*_t_ scatter during fracture behaviour.
(3a)$$\sigma_{N}=\left(\mu_{f}\pm 2\sigma_{f}\right)\cdot \frac{1}{\sqrt{1+\frac{a}{\pi \cdot G}}}=\left(\mu_{f}\pm 2\sigma_{f}\right)\cdot \eta \left(a,\;G\right)$$

Similarly, Equation (2b) can be rewritten as follows.
(3b)$$\sigma_{N}=\left(\mu_{k}\pm 2\sigma_{k}\right)\cdot \frac{1}{2\sqrt{a+\pi \cdot G}}=\left(\mu_{k}\pm 2\sigma_{k}\right)\cdot L_{e}\left(a,\;G\right)$$

The problem of the most existing probabilistic models is that they do not consider the scatters of *f*_t_, *K*_IC_ and fracture strength*σ_N_* prior to experimental measurements [4]. After determinations of the corresponding mean and standard deviation, three *σ*_N_–*η*(*a*, *G*) linear relations from *f*_t_ distribution and three *σ*_N_–*L*_e_(*a*, *G*) straight lines based on *K*_IC_ distribution, can be plotted together including the mean line and upper and lower bounds indicating both *f*_t_ and *K*_IC_ scatters.

In Figure 5, Figure 6, Figure 7, Figure 8 and Figure 9, we take the five ceramic-fracture cases as examples to illustrate the combined application of the non-LEFM model Equation (3) and the normal distribution, and to check the validity and reliability of the *C* = π. For each group of ceramic fracture, *f*_t_ normal distribution illustrated in Figure 5a–Figure 9a and *K*_IC_ normal distribution in Figure 5b–Figure 9b are evaluated from the same experimental measurements (*σ*_N_, *a*). Then the three predicted linear relations from Equation (3a) using *μ*_f_ and *σ*_f_ of *f*_t_ distribution are shown in Figure 5c–Figure 9c, and the three predicted straight lines from Equation (3b) with the *μ_k_* and *μ_k_* ± 2*σ_k_* from *K*_IC_ distribution are illustrated in Figure 5d–Figure 9d. 

Obviously, both *f*_t_ and *K*_IC_ with 96% reliability can be successfully deduced from the seemingly randomly varied experimental data points (*σ*_N_, *a*). It can also be seen that the middle straight lines indicating the mean values of *f*_t_ or *K*_IC_ are almost the same as the corresponding fitted curves respectively. As we all know, the commonly used curve fitting methodology cannot describe the experimental scatter of fracture behaviour, but the proposed model can provide more scientific descriptions. This adds a significant understanding for the fracture of ceramic with and without small defects. Similarly, the scatters of fracture strength *σ*_N_ due to the stochastic characteristic of microstructure and the activation of various fracture mechanisms during the crack-microstructure interaction cannot be identified by the common curve fitting methodology.

Because the *μ*_f_ and *σ*_f_ values are constants, the corresponding slopes of the three straight lines of *σ*_N_–*η*(*a*, *G*) relation indicating mean and upper and lower bounds are kept as constants as *η*(*a*, *G*) increases in Figure 5c–Figure 9c, leading to more *σ*_N_ scatter, which matches very well the experimental results reported in the literature [27,28]. This indicates that a specimen with higher *η*(*a*, *G*) value will have larger fluctuation of *σ*_N_ during fracture. That is larger specimen has larger *σ*_N_ scatter which is absolutely reasonable in physics concept. Here, it should be noted that the percentage of *σ*_N_ variation keeps constant in theory for various specimens with different *η*(*a*_0_, *G*) as the mean and standard deviation are constant. The same conclusions can also be observed from the corresponding results obtained from *K*_IC_ distribution in Figure 5d–Figure 9d. 

It should be noted that the two linear relations of *σ*_N_–*η*(*a*, *G*) and *σ*_N_–*L*_e_ (*a*, *G*) pass through their respective origins, which are fixed points from perspectives of mathematics and physics. During the process of application, this is very useful to helping determine the slopes of the linear relations, which indicate the *f*_t_ or *K*_IC_ values. In addition, it can be seen that *K*_IC_ normal distribution is better than the *f*_t_ normal distribution from the histograms for Sialon, SiC and Al_2_O_3_ cases, and *K*_IC_ and *f*_t_ normal distribution are similar for the other two cases. Thus, it is reasonable that the determination of *C* value (=π) in Section 2 thought more of the *K*_IC_ analyses. 

## 4. Probabilistic Prediction of Ceramic Facture

By employing the two concepts of dimensionless *η*(*a*, *G*) and equivalent length *L*_e_ (*a*, *G*), the nonlinear relation in Equation (1) for assessing the fracture strength of ceramic specimen is linearized in relation as listed in Equation (2). The non-LEFM model combined with the common normal distribution can easily deduce ceramic’s *f*_t_ and *K*_IC_ with a specified reliability from the seemingly randomly varied fracture measurements (*σ*_N_, *a*) as shown in Equation (3).

In fact, the linearized non-LEFM model with a specified reliability or Equation (3) can be conveniently transformed to an application model in nor-linear relation of *σ*_N_–*a* for finally predicting the fracture strength *σ*_N_. If corresponding mean and standard deviation of material strength *f*_t_ or fracture toughness *K*_IC_ are available, the fracture strength of notched specimens with and without small defects can be easily predicted with a specified reliability through Equation (3). Using the values of *f*_t_ property obtained and shown in Figure 5a–Figure 9a, here we take Equation (3a) as an example to show how to apply the model in nonlinear relation for predicting *σ*_N_ with a specified reliability.

The three predicted *σ*_N_–*a* curves with mean and upper and lower bounds are shown in Figure 10 together with the experimental data points. For comparison, the fitted curves for the nonlinear relationship of *σ*_N_–*a* using Equation (3a) are also added to the figure. A direct comparison between the experimental results and the predicted curves shows that the non-LEFM model or Equation (3a) can reliably predict the fracture of notch ceramics as the predicted curves with 96% reliability cover all the data points. In addition, the middle curve indicating the mean of *f*_t_ value is more appropriate than the best fitted curve in each case. More importantly, those *σ*_N_ variations are inevitable for ceramic fracture. The proposed model can describe the *σ*_N_ scatter with a specified reliability, which is beyond the job of common curve fitting. 

## 5. Discussion 

In most of the existing probabilistic models, the fracture measurements data (*a*,*σ_N_*) is usually first obtained from experiment and then fitted by linearized formula to get *f*_t_ and *K*_IC_ values. The problem of the most existing models is that they do not allow the scatters of *f*_t_ and *K*_IC_ data to be predicted before the experimental testing. As we all known, it is easy to establish a normal distribution for a group of data (e.g., *f*_t_ or *K*_IC_ values) with a specified reliability if a group of samples without defects or cracks are tested. But such a practical application soon becomes impractical for a wide crack range from the *f*_t_-controlled fracture over non-LEFM to finally the *K*_IC_-controlled fracture. The non-LEFM model in Equation (3) removes the need of curve fitting for fracture measurements data, and considers the scatters of mechanical properties *f*_t_ and *K*_IC_ through the upper and lower bounds with 96% reliability. 

In the model, the microstructure characteristic *G* of ceramic is explicitly linked to both *f*_t_ and *K*_IC_. In the present study, the five ceramics, Sialon, SiC, Si_3_N_4_ with *G* = 3 μm, Si_3_N_4_ with *G* = 4 μm and Al_2_O_3_, were considered, and they have hugely different grain size *G* ranged from 2 to 20 μm. Since *a**_ch_ = 0.25(*K*_IC_/*f*_t_)^2^, they also have different characteristic crack size *a**_ch_ values due to obvious difference in material properties *f*_t_ and *K*_IC_. Most recently, we studied the relative characteristic crack size *C* = *a**_ch_/*G* at an interval of 0.25 from 2 to 4.5, and found *C* ≥ 3.0 [7,10]. Following the conclusion, the present study further found the *C* = π (or *a**_ch_ = π⋅*G*) is appropriate to consider the effect of microstructure characteristic *G* on the *a**_ch_ value indicating the transition from *f*_t_-controlled fracture to *K*_IC_-controlled fracture.

Equation (3) can be easily used in either linear relations [*σ*_N_–*η*(*a*, *G*), *σ*_N_–*L*_e_ (*a*, *G*)] or nonlinear relations (*σ*_N_–*a*) as required. It should be noted that the predicted results (e.g., *f*_t_, *K*_IC_ and *σ_N_*) with 96% reliability do not change with the different forms of non-LEFM model, which makes the combined use of the proposed model and normal distribution more meaningful and easier in practical applications and data analysis. Furthermore, the *σ_N_*–*η*(*a*, *G*) and *σ_N_*–*L*(*a*, *G*) linear relations through the respective origins make the non-LEFM model easier to determine *f*_t_ and *K*_IC_ values as the origin is absolutely fixed. 

Substituting Equation (1a) into Equation (1b), the mechanical properties *f*_t_ and *K*_IC_ can be linked together through the average grain size *G* and the relative characteristic crack size *C* = π as shown in Equation (4). In another word, the microstructure characteristic *G* has direct influence on *f*_t_ and *K*_IC_ of ceramic.
(4)$$K_{IC}=f_{t}\cdot 2\sqrt{\pi \cdot G}$$

To have a better understanding, the *σ*_N_–*η*(*a*, *G*) linear relations in Figure 5c–Figure 9c were plotted together to show the influnce of microstructure characteristic *G* on *f*_t_ value. To make it clearer, three different ceramics with different *G* are shown in Figure 11a, and two types of Si_3_N_4_ ceramic with slightly different *G* are illustrated in Figure 11b in which only mean lines are given because the experimental points get close each other due to that the two materials are Si_3_N_4_ and have similar *G* values. Obviously, both *f*_t_ and *K*_IC_ have been explicitly linked together through the average grain size, in which the fracture toughness *K*_IC_ is determined from Equation (4) based on the *f*_t_ value and *G* value. Grain size *G* has significant influence on *f*_t_ and *K*_IC_, and can explicitly link the two properties. 

In this study, five groups of fracture data on ceramics with average grain size *G* from 2 to 20 μm have been analyzed. Using the fracture data, it has been proven the proposed non-LEFM model and normal distribution can be combined used to (i) deduce both *f*_t_ and *K*_IC_ with a specified reliability from the seeming randomly varied fracture measurement and (ii) probabilistic predict *σ*_N_ of ceramic specimens with and without defects if *f*_t_ or *K*_IC_ values (corresponding mean and standard deviation) are available. The predictions including mean and upper and lower bounds can be either linear relation as shown in Figure 5, Figure 6, Figure 7, Figure 8 and Figure 9, or nonlinear relation in Figure 10 as required.

## 6. Conclusions

A non-LEFM model for ceramic fracture is proposed and combined with normal distribution, which is applicable to predictions of *f*_t_, *K*_IC_ and *σ*_N_ with consideration of their scatters. The relative characteristic crack size *C* (= *a**_ch_/*G*) indicating the transition from *f*_t_- to *K*_IC_- controlled fracture was determined based on the five groups of fracture data on ceramics with different G from 2 to 20 μm reported in the literature and the dominant fracture mechanism of brittle materials. The main conclusions are as follows: (1)The proposed non-LEFM model combined with normal distribution can be conveniently used either in linear relation to deduce ceramic’s *f*_t_ and *K*_IC_ with a specified reliability from seemingly randomly varied fracture data (σ_N_, *a*) shown in Equation (2), or in nonlinear relation to predict fracture strength *σ_N_* including upper and lower bounds if *f*_t_ and *K*_IC_ ranges are available shown in Equation (2).(2)From perspectives of mathematics and physics, the relative characteristic crack size *C* = *a**_ch_/*G* = π independent of the grain size *G* is determined based on the fracture measurements.(3)Basic mechanical properties of ceramic, *f*_t_ and *K*_IC_, have been linked to the microstructure characteristic *G* and the relative characteristic crack size *C* = π shown in Equation (2), and *f*_t_ and *K*_IC_ are explicitly linked together through the *G* and *C* = π in Equation (4).(4)The upper and lower bounds with 96% reliability for predicting *f*_t_, *K*_IC_ and *σ_N_* scatters in the present study can provide effective insight to design application, which is beyond the job of the commonly used curve fitting.

Although the quantification is specific to fine-grained ceramics, the proposed non-LEFM model and normal distribution analysed approach can also be applied to other polycrystalline solids such as rock.

## Figures and Tables

**Figure 1 materials-12-00727-f001:**
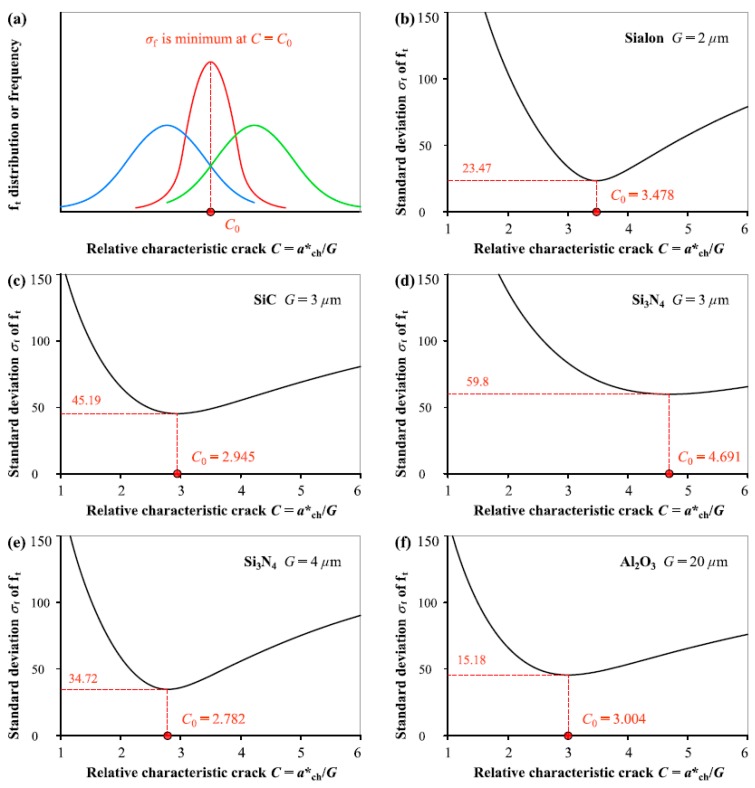
(**a**) Schematic diagram showing variation of *f*_t_ distribution or standard deviation *σ*_f_ with *C* (= *a**_ch_/*G*), and *σ*_f_ as a function of *C* determined by normal distribution for: (**b**) Sialon, (**c**) SiC, (**d**) Si_3_N_4_ with *G* = 3 μm, (**e**) Si_3_N_4_ with *G* = 4 μm, (**f**) Al_2_O_3_.

**Figure 2 materials-12-00727-f002:**
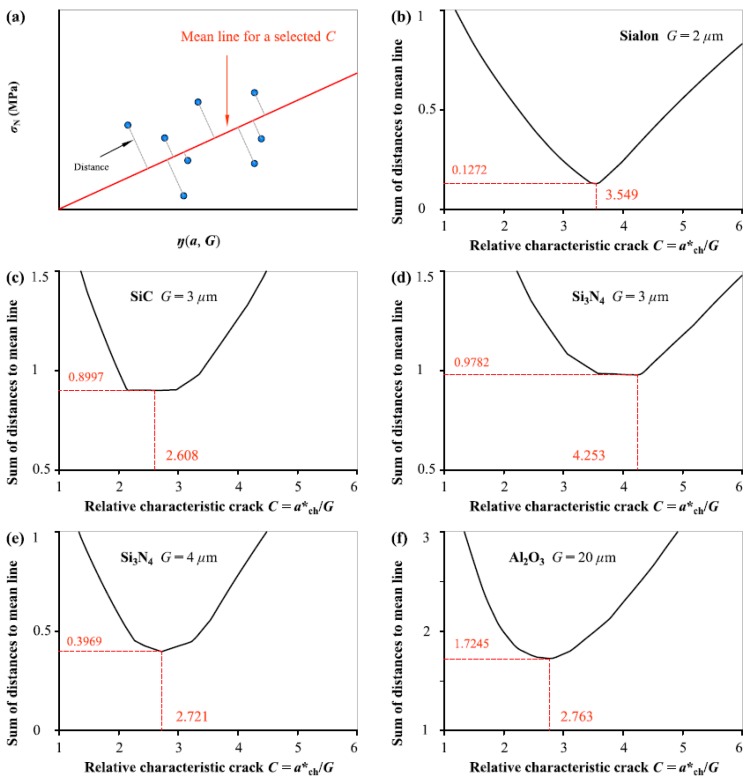
(**a**) Schematic diagram showing the vertical distance of experimental data to the straight line indicating the mean of *f*_t_ values for a selected *C*, and the sum of the distances of all points to the straight line as a function of *C* for: (**b**) Sialon, (**c**) SiC, (**d**) Si_3_N_4_ with *G* = 3 μm, (**e**) Si_3_N_4_ with *G* = 4 μm, (**f**) Al_2_O_3_.

**Figure 3 materials-12-00727-f003:**
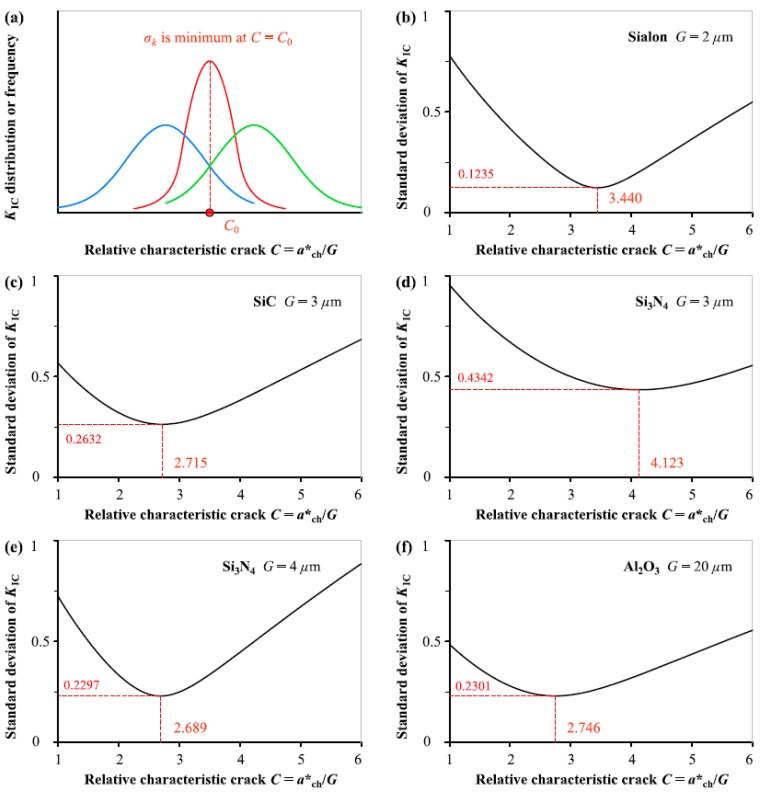
(**a**) Schematic diagram showing variation of *K*_IC_ distribution or standard deviation *σ*_k_ with *C* (= *a**_ch_/*G*), and *σ*_k_ as a function of *C* determined by normal distribution for: (**b**) Sialon, (**c**) SiC, (**d**) Si_3_N_4_ with *G* = 3 μm, (**e**) Si_3_N_4_ with *G* = 4 μm, (f) Al_2_O_3_.

**Figure 4 materials-12-00727-f004:**
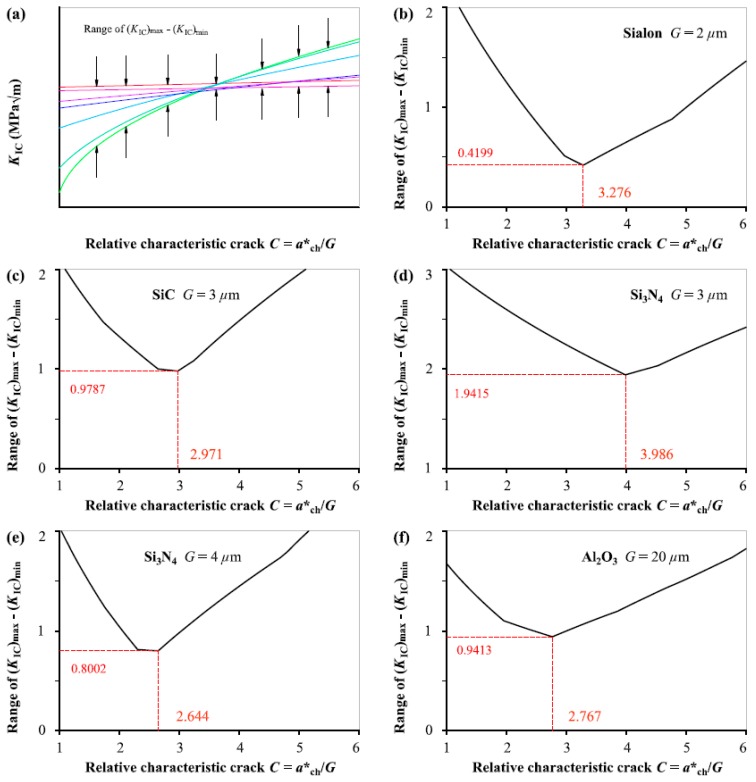
(**a**) Schematic diagram showing variation of the range of (*K*_IC_)_max_–(*K*_IC_)_min_ with *C* (=*a**_ch_/*G*) for: (**b**) Sialon, (**c**) SiC, (**d**) Si_3_N_4_ with *G* = 3 μm, (**e**) Si_3_N_4_ with *G* = 4 μm, (**f**) Al_2_O_3_.

**Figure 5 materials-12-00727-f005:**
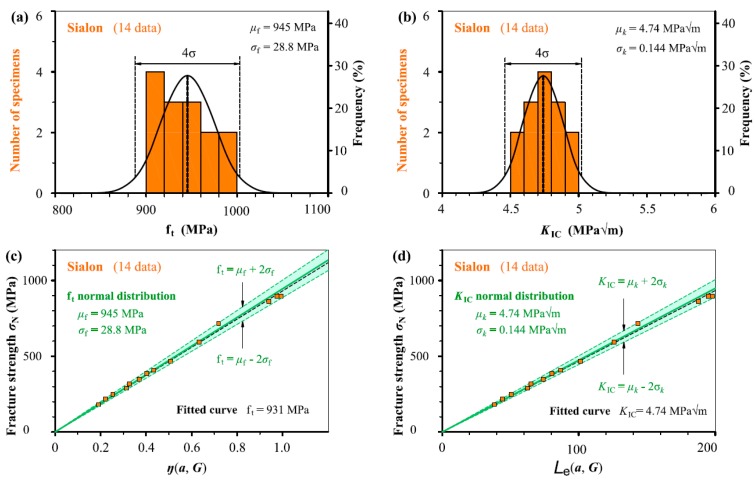
(**a**) *f*_t_ normal distribution and (**b**) *K*_IC_ normal distribution evaluated from the same fracture measurements (*σ*_N_, *a*) of Sialon. Three predicted straight lines indicating the mean and upper and lower bounds (**c**) from Equation (3a) and (**d**) from Equation (3b) using the analysed results listed in (**a**) and (**b**) respectively together with the experimental data points.

**Figure 6 materials-12-00727-f006:**
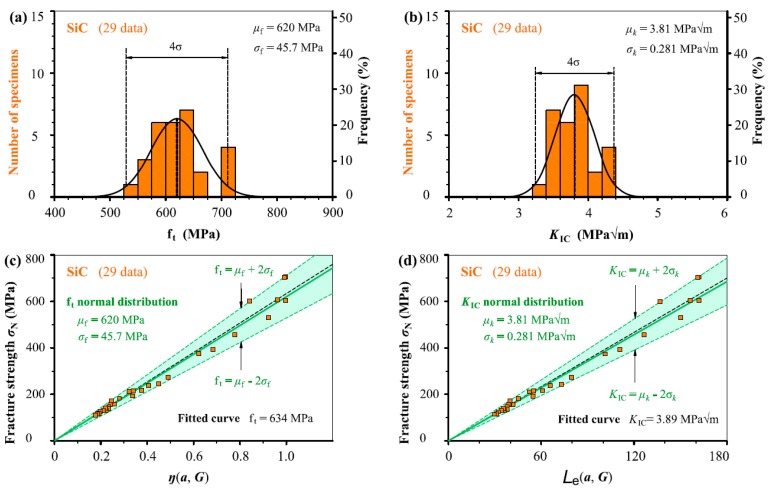
(**a**) *f*_t_ normal distribution and (**b**) *K*_IC_ normal distribution evaluated from the same fracture measurements (*σ*_N_, *a*) of SiC. Three predicted straight lines indicating the mean and upper and lower bounds (**c**) from Equation (3a) and (**d**) from Equation (3b) using the analysed results listed in (**a**) and (**b**) respectively together with the experimental data points.

**Figure 7 materials-12-00727-f007:**
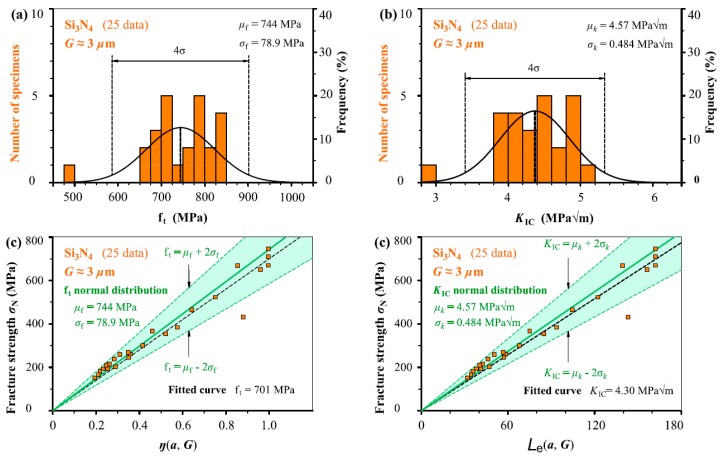
(**a**) *f*_t_ normal distribution and (**b**) *K*_IC_ normal distribution evaluated from the same fracture measurements (*σ*_N_, *a*) of Si_3_N_4_ with *G* = 3 μm. Three predicted straight lines indicating the mean and upper and lower bounds (**c**) from Equation (3a) and (**d**) from Equation (3b) using the analysed results listed in (**a**) and (**b**) respectively together with the experimental data points.

**Figure 8 materials-12-00727-f008:**
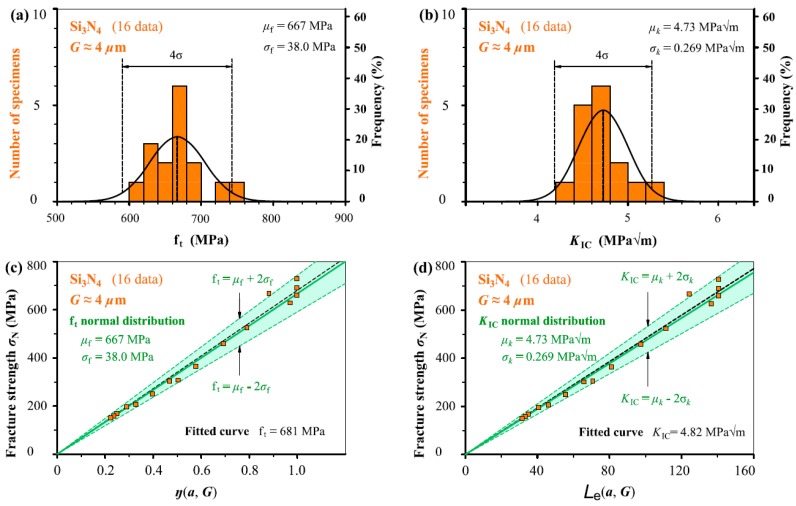
(**a**) *f*_t_ normal distribution and (**b**) *K*_IC_ normal distribution evaluated from the same fracture measurements (*σ*_N_, *a*) of Si_3_N_4_ with *G* = 4 μm. Three predicted straight lines indicating the mean and upper and lower bounds (**c**) from Equation (3a) and (**d**) from Equation (3b) using the analysed results listed in (**a**) and (**b**) respectively together with the experimental data points.

**Figure 9 materials-12-00727-f009:**
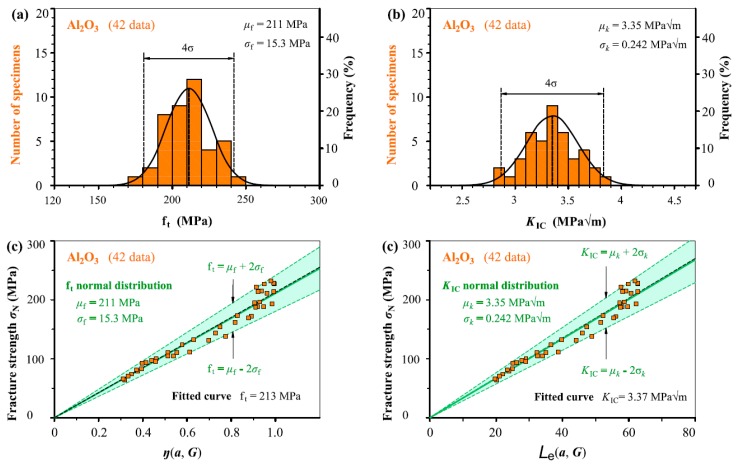
(**a**) *f*_t_ normal distribution and (**b**) *K*_IC_ normal distribution evaluated from the same fracture measurements (*σ*_N_, *a*) of Al_2_O_3_. Three predicted straight lines indicating the mean and upper and lower bounds (**c**) from Equation (3a) and (**d**) from Equation (3b) using the analysed results listed in (**a**) and (**b**) respectively together with the experimental data points.

**Figure 10 materials-12-00727-f010:**
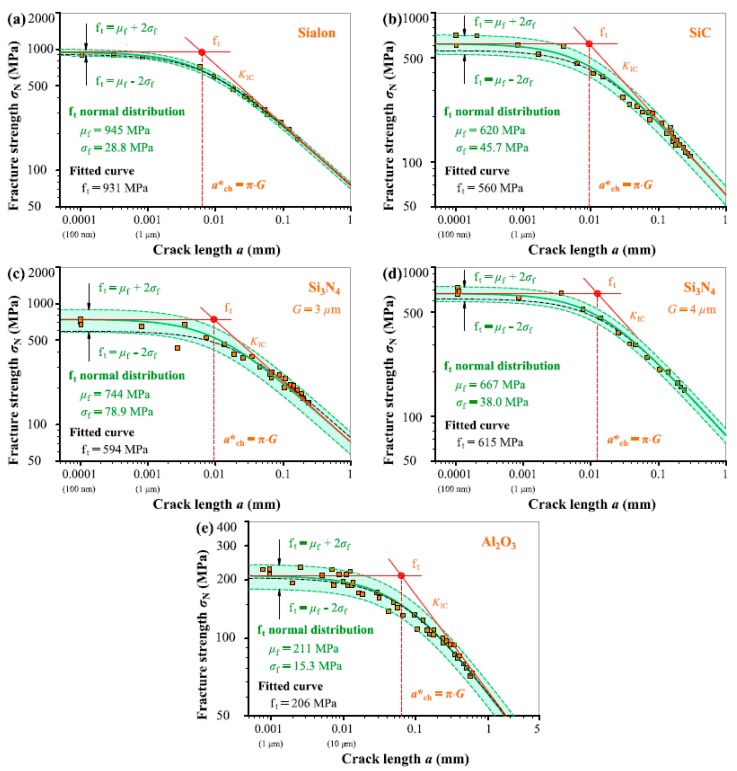
Fracture predictions using Equation (3a) with the *μ*_f_ and *σ*_f_ values from *f*_t_ normal distribution, including mean and upper and lower bounds with 96% reliability together with the experimental data points (*σ*_N_, *a*) and fitted curve for: (**a**) Sialon, (**b**) SiC, (**c**) Si_3_N_4_ with *G* = 3 μm, (**d**) Si_3_N_4_ with *G* = 4 μm, and (**e**) Al_2_O_3_.

**Figure 11 materials-12-00727-f011:**
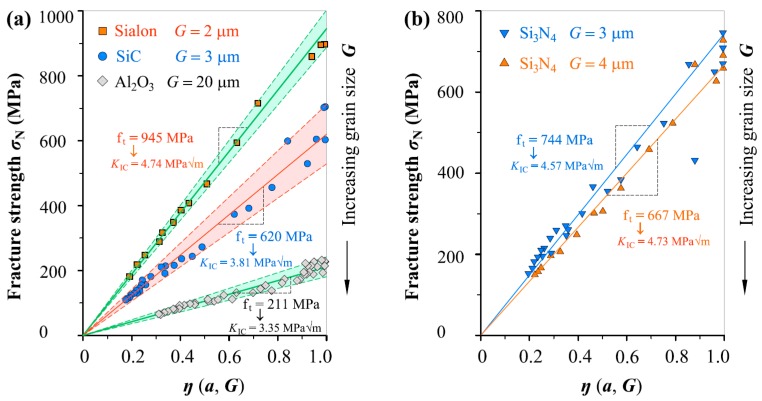
Comparison of different ceramic fracture indicating the *G* influnce on *f*_t_ and *K*_IC_: (**a**) Sialon, SiC and Al_2_O_3_, (**b**) Si_3_N_4_ with *G* = 3 μm and *G* = 4 μm.

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
