# Peer review of "Probabilistic Prediction of Strength and Fracture Toughness Scatters for Ceramics Using Normal Distribution"

_materials, 2019, doi:10.3390/ma12050727_

Round 1
Reviewer 1 Report
The paper concerns an important aspect of determination of mechanical properties of polycrystalline materials. Presented idea is quite clearly presented. I have only one restriction. Authors completly skipped the problem of porosity in sintered polycrystals. They not presented any image of microstructure of investigated materials, so readers have to believe that materials were completly dense. Of cours I understand that authors assumed that residual pores were small and did not influence strength or fracture toughness but it was very risky to assumed thet pores could be ignored completly. My question is: "Is there any in place for porosity influence on mechanical properies of materials in presented model? If the answer will be "not" please explain your point more clearly in the introduction section.
Author Response
Many thanks for the Reviewer’s professional comment and suggestion. The variations in material properties such as ft & KIC and fracture strength, are mainly due to the heterogeneity of microstructure, defects like pores and the activation of various fracture mechanisms during the crack-microstructure interactions. The proposed probabilistic model shown in Eq. (3) is to consider their variations or scatters specified by both mean value and standard deviation. We have explained it in the revised manuscript according to the Reviewer’s suggestion.

Reviewer 2 Report
In general the paper will be a valuable contribution to the fracture mechanics analysis for ceramics.
1. I guess that the notion "the characteristic crack" is used wrong in the paper because the sense meant is the length or size of that crack. So, I propose to change "the characteristic crack" by "the characteristic crack size" in many places in the paper.
2. The authors frequently use misprint "um" instead of "μm"
3. The word "indenpent" is not officially recognized in English.
Comments concerned to the highlights in the file attached.
Page 1, line 13
Apparently, should be "crack size a*ch"
Page 1, line 15
"reference"? - Apparently, the authors mean "available in the reference list" but not their own experiment.
Page 2, line 64
there is no explanation what is denoted by "a".
Page 2, line 80
What is denoted by "W"?
Page 3, lines 105-106
The sentence seems incomplete (unfinished).
Page 4, lines 122-123
I would say these are not "vertical distances" but 'distances by normal to the mean line'.
Page 5, Fig. 2
The notion η(a,G) is used before it is introduced in Eq. (2a) at page 8.
Page 7, line 157
I can't agree that the defects could be regarded as a mechaism of fracture mechancs (LEFM mechanism).
Page 7, line 158
An example of correct usage of explanation for a*ch as "specified crack length" and not the crack itself.
Page 7, line 159
Incorrect usage of the word "microstructure" -> "microstructure characteristic".
Page 9, lines 217-218
Not for all cases (e.g. Al2O3). See the file attached.
Page 13, line 283
There is no number indicating reference in square brackets [].

Author Response
Reviewer 2
In general the paper will be a valuable contribution to the fracture mechanics analysis for ceramics.
1. I guess that the notion "the characteristic crack" is used wrong in the paper because the sense meant is the length or size of that crack. So, I propose to change "the characteristic crack" by "the characteristic crack size" in many places in the paper.
Thanks the Reviewer! The characteristic crack has been changed to the characteristic crack size in the revised manuscript.
2. The authors frequently use misprint "um" instead of "μm"
All the misprints have been corrected in the revised manuscript.
3. The word "indenpent" is not officially recognized in English.
Yes! The “indenpent of” has been replaced by “not dependent on”.
4. Comments concerned to the highlights in the file attached.
Many thanks the Reviewer! All have been corrected accordingly.
5. Page 1, line 13. Apparently, should be "crack size a*ch"
Yes, characteristic crack size a*ch.
6. Page 1, line 15. "reference"? - Apparently, the authors mean "available in the reference list" but not their own experiment.
Yes! All the data was from the reference (Usami S, et al., Eng Fract Mech 1986, 23: 745), and has been added in the revised manuscript.
7. Page 2, line 64, there is no explanation what is denoted by "a".
“a” denotes the initial crack size.
8. Page 2, line 80. What is denoted by "W"?
“W” denotes the beam/specimen height.
9. Page 3, lines 105-106. The sentence seems incomplete (unfinished).
It has been corrected as follows: The largest number of experimental data points selected in this analysis is 42 from Al2O3 fracture, and the minimum number is mere 14 from Sialon.
10. Page 4, lines 122-123. I would say these are not "vertical distances" but 'distances by normal to the mean line'.
It has been corrected in the revised manuscript.
11. Page 5, Fig. 2. The notion η(a,G) is used before it is introduced in Eq. (2a) at page 8.
The notion of η(a,G) is explicitly proposed in Eq. (2a0. In Fig. 2, a group of ft values can be obtained using Eq. (1a) based on the fracture measurements (a, sN) for a giving C value. The sum of the distances of all the data points to the mean line indicating the mean value of ft can be calculated accordingly.
12. Page 7, line 157. I can't agree that the defects could be regarded as a mechanism of fracture mechanics (LEFM mechanism).
I agree with the Review. The defects in solids are not the LEFM mechanism. The corresponding content has been changed as follows:
Furthermore, the dominant non-LEFM mechanism of brittle materials such as ceramic should have relation to the defects in solids, which may be linked to microstructure characteristic G.
13. Page 7, line 158. An example of correct usage of explanation for a*ch as "specified crack length" and not the crack itself.
Thanks the Review!
14. Page 7, line 159. Incorrect usage of the word "microstructure" -> "microstructure characteristic".
The “microstructure” has been changed to “microstructure characteristic”.
15. Page 9, lines 217-218. Not for all cases (e.g. Al2O3). See the file attached.
Many thanks! We have corrected accordingly in the revised manuscript.
16. Page 13, line 283. There is no number indicating reference in square brackets [].
We apologize for the mistake. It has been corrected in the revised manuscript.